Quantification of antibiotic resistance genes and mobile genetic in dairy manure

http://orcid.org/0000-0003-2830-4754 Wang Yi 1
Pandey Pramod 1 pkpandey@ucdavis.edu
Chiu Colleen 1
Jeannotte Richard 1 2
Kuppu Sundaram 1
Zhang Ruihong 3
http://orcid.org/0000-0003-2028-8761 Pereira Richard 1
http://orcid.org/0000-0002-7471-1978 Weimer Bart C. 1
Nitin Nitin 4
http://orcid.org/0000-0003-0330-5013 Aly Sharif S. 1 5
1 Department of Population Health and Reproduction, University of California, Davis , Davis, California , United States
2 Department of Plant Science, University of California, Davis , Davis, California , United States
3 Department of Biological and Agricultural Engineering, University of California, Davis , Davis, California , United States
4 Department of Food Science and Technology, University of California, Davis , Davis, California , United States
5 Veterinary Medicine Teaching and Research Center, School of Veterinary Medicine, University of California, Davis , Davis, California , United States
Czajkowski Robert
Electronic publication date: 2021 Dec 23
Publication date: 2021
Volume: 9
Electronic Location ID: e12408
Received 2020 Dec 23; Accepted 2021 Oct 8
Copyright: © 2021 Wang et al.
Copyright year: 2021
Copyright holder: Wang et al.
License: This is an open access article distributed under the terms of the Creative Commons Attribution License, which permits unrestricted use, distribution, reproduction and adaptation in any medium and for any purpose provided that it is properly attributed. For attribution, the original author(s), title, publication source (PeerJ) and either DOI or URL of the article must be cited.
License URL: https://creativecommons.org/licenses/by/4.0/

Keywords: Antibiotic resistance genes, Mobile genetic elements, Dairy manure, Real-time pcr, Gene sequencing

Funding: USDA National Institute of Food and Agriculture through Center for Food Animal Health (CFAH) Animal Health Project CALV-AH-367 University of California-Davis, Davis, California This work was supported by the USDA National Institute of Food and Agriculture through Center for Food Animal Health (CFAH), Animal Health Project (Accession Number CALV-AH-367), University of California-Davis, Davis, California. The funders had no role in study design, data collection and analysis, decision to publish, or preparation of the manuscript.

==============================
Background

Antibiotic resistance genes (ARGs) are considered to be emerging environmental contaminants of concern potentially posing risks to human and animal health, and this research studied the prevalence of antimicrobial resistance in dairy manure.

Methods

This study is focused on investigating prevalence of ARGs in California dairy farm manure under current common different manure management. A total of 33 manure samples were collected from multiple manure treatment conditions: (1) flushed manure (FM), (2) fresh pile (FP), (3) compost pile (CP), (4) primary lagoon (PL), and (5) secondary lagoon (SL). After DNA extraction, all fecal samples were screened by PCR for the presence of eight ARGs: four sulfonamide ARGs (sulI, sulII, sulIII, sulA), two tetracycline ARGs (tetW, tetO), two macrolide-lincosamide-streptogramin B (MLSB) ARGs (ermB, ermF). Samples were also screened for two mobile genetic elements (MGEs) (intI1, tnpA), which are responsible for dissemination of ARGs. Quantitative PCR was then used to screen all samples for five ARGs (sulII, tetW, ermF, tnpA and intI1).

Results

Prevalence of genes varied among sample types, but all genes were detectable in different manure types. Results showed that liquid-solid separation, piling, and lagoon conditions had limited effects on reducing ARGs and MGEs, and the effect was only found significant on tetW (p = 0.01). Besides, network analysis indicated that sulII was associated with tnpA (p < 0.05), and Psychrobacter and Pseudomonas as opportunistic human pathogens, were potential ARG/MGE hosts (p < 0.05). This research indicated current different manure management practices in California dairy farms has limited effects on reducing ARGs and MGEs. Improvement of different manure management in dairy farms is thus important to mitigate dissemination of ARGs into the environment.

Introduction

Antibiotic resistance is an emerging concern to public health (CDC, 2013; Frieri, Kumar & Boutin, 2017; Pang et al., 2019; Zaman et al., 2017). In the United States of America, it is estimated that more than 2.8 million people are possibly infected by antibiotic-resistant bacteria each year (CDC, 2019). The total economic cost to the U.S. economy is estimated up to $55 billion a year due to lost wages, extended hospital stays, and premature deaths (CDC, 2013; Roberts et al., 2009). The use of antibiotics in animal husbandry is one of the leading factors causing widespread antibiotic resistance (CDC, 2019). Every year, approximately 80% of all antimicrobial drugs are applied in animal farming to treat/prevent infectious diseases (FDA, 2012; FDA, 2019). Out of this, approximately 70% is used for non-therapeutic purposes (UCS, 2001). The elevated quantity of antibiotic residue in fecal matter is potentially due to low absorption in the cattle body (Jjemba, 2002).

Dairy manure is used as fertilizers in cropland, and potential impacts of manure borne antibiotic resistance genes (ARGs) on the environment are yet to be fully understood (Baquero, Martínez & Cantón, 2008; Han et al., 2018; Kumar et al., 2005; Wind et al., 2018; Zhao et al., 2017). Dairy manure is considered potential mediator and reservoir for ARGs (Allen et al., 2010; Guardabassi, Schwarz & Lloyd, 2004). When dairy manures are applied as fertilizer, manure born ARGs and antibiotic residues may transfer into crop land soil, and subsequently to ambient water bodies (Bennett, 2008; Gogarten & Townsend, 2005).

Previous research showed that antibiotic resistance in environmental bacteria is selected by antibiotic residues (Baquero, Martínez & Cantón, 2008; Pruden et al., 2013), and ARGs can proliferate in the host bacteria and transfer to other microbes, including human pathogens, through horizontal gene transfers (HGTs) (Bennett, 2008; Gogarten & Townsend, 2005).

In terms of quantity, more than 369 million tons of manure are produced in the USA annually (USDA, 2012), and the majority of this dairy manure is used as fertilizer in various cropland. Currently, California is the top milk producing state in the United States of America, and dairy farms also produces around 60 million tons of manure annually (USDA, 2016).

While managing manure in dairy farms, the flushed system is one of the most commonly adopted methods for manure handling and management in dairy farms in California. Flush system has many benefits, including low labor, ease of handling, and reduced operating cost (CARB, 2017; Kaffka et al., 2016; Meyer et al., 2011), however, it also produces enormous amount of liquid manure. Identifying the suitable treatment methods, which can reduce the contamination levels in manure is important for food safety. Therefore, this study was focused on studying the prevalence of ARGs in the flush manure management system. In a flushed system (Fig. 1), a dairy barn is flushed with recycled water from a lagoon system which holds liquid manure and then flushed manure passes through a liquid-solid separator, where it is separated into solid and liquid waste streams. Solid manure is piled, and in some cases, it is composted in dairy farms before being applied into cropland as fertilizers.

Figure 1 Typical processing of flushed manure in dairy farms in Central Valley California.

Source credit: Pandey et al. (2018).

Previous reports emphasize the importance of understanding the fate of ARGs in livestock manure treatments (Flores-Orozco et al., 2020; Gou et al., 2018; Howes, 2017; Ma et al., 2018). The abundance of ARGs in livestock waste varies among farm types and locations (He et al., 2020). McKinney et al. (2010) examined the behavior of ARGs in eight livestock lagoon systems. Hurst et al. (2019) studied the abundance of 13 ARGs in untreated manure blend pits. The authors found ARGs abundance varied among farms, and ARG concentrations generally did not correlate to average antimicrobial usage due to complex environmental factors (Flores-Orozco et al., 2020; Huang et al., 2019; Pei et al., 2007; Selvam et al., 2012; Sun et al., 2016; Wang et al., 2012).

Previous findings of an investigation in three pig farm wastewater treatment systems in China showed a relative abundance, and most ARGs were significantly higher in wastewater lagoon than in fresh manures even after treatment (Cheng et al., 2013). Though these studies do provide preliminary understanding, knowledge about the ARGs in California flushed system in dairy farm is yet to be understood.

While issues of antibiotics in environment are considered important, standard guidelines and practices capable of reducing antibiotic resistance are yet to be proposed (Allen et al., 2010 D’Costa et al., 2006; Ghosh & LaPara, 2007). Currently, knowledge of species and antibiotic resistance profile of unculturable bacteria is lacking. The use of culture-independent methods, such as polymerase chain reaction (PCR) and quantitative polymerase chain reaction (qPCR) are preferred due to simplicity and fast results and have the potential to produce relatively more comprehensive and reproducible knowledge of ARG profiles.

Considering the importance of animal-agricultural system in food supply, and associated by products, understanding the role of various manure management practices in reducing manure borne contamination is crucial for both public health and environmental health. Human health is closely connected to animal health and their shared environment (One Health); therefore, it is important to explore antimicrobial resistance genes content of manure and the optimal management practices to reduce ARGs content for reducing unwanted potential impacts on reduced efficacy of antimicrobial drugs in clinical practice. The aim of this study was to investigate the fate of ARGs and MGEs in manure under the flushed manure management system. The specific objectives of this research were: (1) estimate the prevalence of antibiotic resistance genes in manure under different manure management practices; (2) quantify ARGs and MGEs in dairy manure; and (3) determine the relationships among ARGs, MGEs, and microbial communities.

Materials & Methods

Solid and liquid manure sampling in dairy farms

In dairy farms, liquid and solid dairy manure samples were collected in California Central Valley from multiple dairy farms (Pandey et al., 2018). A total of 33 solid/liquid manure samples were collected from Tulare, Glenn, and Merced counties in California Central Valley. California Central Valley has approximately 91 percent of dairy cows and over 80 percent of dairy facilities in California (CARB, 2017). In dairy farms, solid manure samples were collected from fresh/old piles (0 to 6 months old) (n = 14), and liquid manure samples were collected from flushed manure pits and primary/secondary lagoons (0 to 6 months old storage) (n = 19). The solid manure samples collected from fresh piles (less than 2 weeks old pile) were termed as Fresh Pile (FP). The solid samples collected from old piles were termed as Compost Pile (CP). The studied CP here does not necessarily mean that the piles were maintained under thermophilic temperature and mixing conditions of a standard composting process. Similarly, lagoon system in dairy farms were not necessarily under strict anaerobic environments. The liquid manure samples collected from flushed manure pit were termed as Flushed Manure (FM), while the liquid manure samples collected from primary lagoons and secondary lagoons were termed as Primary Lagoon (PL) and Secondary Lagoon (SL), respectively. In each dairy facility, one L of liquid manure sample from each pond, and approximately 500–600 g of solid manure from each pile were collected in sterile bottles. Samples were then transported on wet ice after collection and stored at −20 °C before DNA extraction.

DNA extraction from manure samples

In manure samples, genomic DNA was extracted either using the MO BIO PowerSoil® DNA Isolation Kit or MO BIO PowerWater® DNA Isolation Kit (MO BIO Laboratories Inc., Venlo, Netherlands). All solid samples and liquid samples with turbid and sludge-like consistency were processed by the MO BIO PowerSoil® kit. For sludge-like liquid samples (sample with high solid), 10 mL of each sample were centrifuged in 50 mL polypropylene tubes at 13,000 rpm for 10 min and 0.25 g of the resulting pellet was used for bead beating. Liquid samples with clear-to-low turbidity were processed by the MO BIO PowerWater® kit, and 10–200 mL of each was filtered through a Millipore filter (0.45-µm pore size). The quality and concentration of the DNA were assessed by NanoDrop 1000 spectrophotometer (Thermo Scientific, Waltham, MA, USA). All extracted DNA from manure samples were stored at −20 °C before PCR amplification.

PCR assays for detection of resistance genes in manure

PCR assays for detection of sul, tet and erm were designed. It was reported sul, tet and erm are three of the most frequently detected ARGs classes in livestock waste, which match the major classes of antibiotics used in animal growth promotion and disease control (He et al., 2020). Primers designed in previous work targeting sul, tet and erm genes were used in this study to amplify ARGs (Garder, Moorman & Soupir, 2014; Hu et al., 2015; Pei et al., 2007), which are shown in Table 1. Subsequently, PCR assays were performed to determine gene detectability in the manure samples. These assays were carried out using the KAPA2G Robust HotStart Ready Mix PCR Kit (KAPA) in a 25 µL volume reaction. The PCR reaction consisted 12.5 µL 2 × KAPA2G Robust Hotstart Ready Mix, 1.25 µL 10 mM each primer, and two µL of the template. The temperature program consisted of initial denaturation at 95 °C, followed by 40 cycles of 15 s at 95 °C; 30 s at the 60 °C (55 °C for tetO, tetW, ermB and ermF); 30 s at 72 °C, and a final extension step for 1 min at 72 °C. Primer and temperature conditions for intI and tnpA genes are presented in Table 1. PCR products were verified by gel electrophoresis. Purification, and cloning was done using the TOPO TA Cloning Kit (Invitrogen, Waltham, MA, USA) according to the manufacturer’s instructions. Subsequently, clones were sequenced to verify the insert of the targeted gene (sequencing and verification results are shown in Figs. S1–S6). Plasmids carrying the target genes were extracted and used as positive controls for qPCR to generate standard curves.

Table 1 Synthetic oligonucleotides used in this study.

Primer	Target gene	Sequences (direction 5′–3′)	Traditional PCR annealing temp (°C)	qPCR annealing temp (°C)	Amplicon size (bp)	Reference	
sulI-FW	sulI	CGCACCGGAAACATCGCTGCAC	60	60	163	(Pei et al., 2006)	
sulI-RV	TGAAGTTCCGCCGCAAGGCTCG	
sulII-FW	sulII	TCCGGTGGAGGCCGGTATCTGG	60	60	191	
sulII-RV	CGGGAATGCCATCTGCCTTGAG	
sulIII-FW	sulIII	TCCGTTCAGCGAATTGGTGCAG	60	60	128	
sulIII-RV	TTCGTTCACGCCTTACACCAGC	
sulA-FW	sulA	TCTTGAGCAAGCACTCCAGCAG	60	60	299	
sulA-RV	TCCAGCCTTAGCAACCACATGG	
tetW-FW	tetW	GAGAGCCTGCTATATGCCAGC	55	53.9	168	(Aminov, Garrigues-Jeanjean & Mackie, 2001)	
tetW-RV	GGGCGTATCCACAATGTTAAC	
tetO-FW	tetO	ACGGARAGTTTATTGTATACC	55	48.5	171	
tetO-RV	TGGCGTATCTATAATGTTGAC	
ermB-FW	ermB	GGTTGCTCTTGCACACTCAAG	55	51.2	191	(Koike et al., 2010)	
ermB-RV	CAGTTGACGATATTCTCGATTG	
ermF-189f	ermF	CGACACAGCTTTGGTTGAAC	55	51.4	309	(Chen et al., 2007)	
ermF-497r	GGACCTACCTCATAGACAAG	
HS463a	intI1	CTGGATTTCGATCACGGCACG	60	55.7	473	(Barlow et al., 2004)	
HS464	ACATGCGTGTAAATCATCGTCG	
tnpA-04F	tnpA-04	CCGATCACGGAAAGCTCAAG	60	56	101	(Zhu et al., 2013)	
tnpA-04R	GGCTCGCATGACTTCGAATC	
357F	16S rRNA gene	CCTACGGGAGGCAGCAG	60	56	193	(Muyzer, De Waal & Uitterlinden, 1993)	
518R	ATTACCGCGGCTGCTGG	

Real-time quantitative PCR (RT-qPCR) analysis in manure

Targeted genes and 16S rRNA gene qPCR reactions were performed using a StepOnePlus™ System (Life Technology, Carlsbad, CA, USA) in a 20 µL of reaction mixture (10 µL PowerUp™ SYBR™ Green Master Mix [2x]) (Life Technology, Carlsbad, CA, USA, two µL of 10 mM each primer, and two µL of template) with a temperature program of 2 min at 50 °C for UDG activation and 2 min at 95 °C for Dual-Lock™ DNA polymerase activation. This was followed by 40 cycles of 15 s at 95 °C; 15 s at 50–60 °C (60 °C for Tm > 60 °C and at Tm for Tm < 60 °C); 1 min at 72 °C. Each reaction was conducted in triplicates.

The average copy and standard deviation were calculated using triplicate for each reaction. Melting curve analysis was used to detect nonspecific amplification. Standard curves were included in each qPCR plate by performing serial 10-fold dilutions of the standards. The efficiency of the PCR was calculated by Efficiency = 10−(1/slope) – 1. All standard curves had a r2 > 0.99 and an amplification efficiency of 90–110%. The detection limit for each gene was determined by the highest dilution that produced a consistent CT value (within 5% deviation). If the standard deviation was more than 5% then two samples with the smallest difference were used for calculation.

The absolute copy number of genes was quantified by referring to the corresponding standard curve obtained by plotting threshold cycles versus log-copy number of genes. Levels of targeted genes were normalized as the percentage of copy number of a gene/copy number of 16S rRNA gene for each sample to emphasize the relative abundance in environmental samples (Alexander et al., 2011; Marti, Jofre & Balcazar, 2013; Selvam et al., 2012).

16S rRNA gene sequencing in manure samples

The high-throughput sequencing for 16S rRNA gene is described elsewhere (Pandey et al., 2018). Sequencing was performed by DNA Technologies Core Facility of the Genome Center at the University of California-Davis using the Illumina MiSeq platform. The V3 and V4 hypervariable region of the 16S rRNA gene was amplified using the forward primer: (5′-TCGTCGGCAGCGTCAGATGTGTATAAGAGACAGCCTACGGGNGGCWGCAG-3′) and the reverse primer: (5′-GTCTCGTGGGCTCGGAGATGTGTATAAGAGACAGGACTACHVGGGTATCTAATCC-3′) (Klindworth et al., 2012). For quality control, barcodes and primers were allowed to have one and four mismatches, respectively. Primer sequence reads were then trimmed, and sequences were merged into a single amplicon sequence using FLASH2. Assignment of sequence to phylotypes was performed in the RDP database using the RDP Bayesian classifier (bootstrap confidence score > 50%). Further, covariates were generated using relative abundance of bacterial taxa in each sample. Stepwise discriminant analysis models built in JMP Pro 13.0 were performed until only variables with a p-value < 0.05 were retained (Pandey et al., 2018).

Data analysis

Statistical analysis on gene abundance data was performed using the published approaches (Burch, Sadowsky & LaPara, 2016; Sandberg & LaPara, 2016; Sun et al., 2016). Data were log-transformed before statistical analysis. Ordinary one-way ANOVA was used to evaluate the influence of dairy manure conditions on gene reductions by GraphPad Prism 8. Residuals were checked by Brown-Forsythe test of heteroscedasticity and Anderson-Darling test of normality (α = 0.05). Tukey’s multiple comparison test was used for comparing gene levels under different conditions (α = 0.05). Multiplicity adjusted p-value was reported for each comparison. A Principal Component Analysis (PCA) plot and hierarchical clustering plot were conducted by MetaboAnalyst 3.0 to find similarity among samples. Correlation networks were created by MetScape 3.1.3 and Cytoscape 3.4.0. CorrelationCalculator 1.0.1 was used based on Debiased Sparse Partial Correlation (DSPC) method to calculate partial correlation values and p-values for each pair in the network. Range for edges was set to partial correlation values (corr. pcor) of <−0.20 or >+0.20.

Results

PCR for gene presence

Firstly, PCR assays were applied to explore whether the gene was detectable or not in each sample. PCR screening results in all 33 samples are shown in Table 2. Four different manure management groups, FP, FM, PL, and SL, had similar positive percentages of gene types. CP group had a significantly lower percentage (p = 0.02), with an average of 47% types of targeted genes. One sample (PL2) was found with no detection. Four samples (FP1, FP5, FP7, and SL4) were found with all ten genes. The most abundant gene was sulII, and it was present in a total of 93.9% among all samples, with a percentage of 92.9% in solid samples and 94.7% in liquid samples. The lowest one, sulA, was positive in a total of 12.1% among all samples, and was detected in 21.4% solid samples and 5.3% liquid samples. Liquid samples normally had a higher percentage of detectable genes, except for sulIII and sulA. sulII, tetW, ermF, tnpA and intI1 were selected for further study to quantify the gene concentrations because of their representation of different antibiotic resistance mechanisms and high prevalence among the samples. Fisher’s exact test for contingency table analysis showed the overall gene detection rate in CP group was significantly lower than FP, FM, PL, and SL (p < 0.01).

Table 2 Detection of resistance gene families in dairy manure.

	Sample ID	sulI	sulII	sulIII	sulA	tetO	tetW	ermB	ermF	tnpA	intI1	Percentage	
1	FP1	+	+	+	+	+	+	+	+	+	+	100%	
2	FP2	+	+	–	–	–	+	–	–	+	+	50%	
3	FP3	+	+	–	–	+	+	–	+	+	–	60%	
4	FP4	+	+	–	–	–	+	+	+	+	+	70%	
5	FP5	+	+	+	+	+	+	+	+	+	+	100%	
6	FP6	+	+	+	–	+	+	+	+	+	+	90%	
7	FP7	+	+	+	+	+	+	+	+	+	+	100%	
8	CP1	+	+	–	–	–	+	–	+	+	+	60%	
9	CP2	+	+	+	–	+	+	–	+	+	–	70%	
10	CP3	–	–	–	–	–	–	–	–	–	+	10%	
11	CP4	+	+	+	–	+	+	+	+	+	+	90%	
12	CP5	–	+	–	–	–	–	–	–	–	+	20%	
13	CP6	+	+	–	–	–	+	–	–	+	–	40%	
14	CP7	–	+	–	–	–	+	–	+	+	–	40%	
15	FM1	+	+	–	–	–	+	+	+	+	+	70%	
16	FM2	+	+	–	–	–	+	+	+	+	+	70%	
17	FM3	+	+	+	–	+	+	+	+	+	+	90%	
18	FM4	+	+	–	–	+	+	+	+	+	+	80%	
19	FM5	+	+	–	–	+	+	+	+	+	+	80%	
20	FM6	+	+	–	–	+	+	+	+	+	+	80%	
21	PL1	+	+	–	–	+	+	+	+	+	+	80%	
22	PL2	–	–	–	–	–	–	–	–	–	–	0	
23	PL3	+	+	+	–	+	+	+	+	+	+	90%	
24	PL4	+	+	+	–	+	+	+	+	+	+	90%	
25	PL5	+	+	+	–	+	+	+	+	+	+	90%	
26	PL6	+	+	–	–	+	+	+	+	+	+	80%	
27	PL7	+	+	–	–	+	+	+	+	+	+	80%	
28	PL8	+	+	+	–	+	+	+	+	+	+	90%	
39	SL1	+	+	–	–	–	+	+	+	+	+	70%	
30	SL2	+	+	+	–	+	+	+	+	+	+	90%	
31	SL3	+	+	+	–	–	+	+	+	+	+	80%	
32	SL4	+	+	+	+	+	+	+	+	+	+	100%	
33	SL5	+	+	–	–	+	+	–	+	+	–	60%	
Positive percentage	87.9%	93.9%	42.4%	12.1%	63.6%	90.9%	69.7%	84.8%	90.9%	81.8%		
Notes:

+: present; −: absent.

FP, Fresh Pile; CP, Compost Pile; FM, Flushed manure; PL, Primary Lagoon; SL, Secondary Lagoon.

Quantification of resistance related genes

Five genes (sulII, tetW, ermF, tnpA and intI1) were quantified by qPCR in 33 dairy manure samples taken from different manure management conditions. DNA templates for qPCR were the same batch of extractions as PCR (Table 1). The numbers of copies of the five resistance related genes quantified at each sample were then normalized to the number of copies of bacterial 16S rRNA gene. Data is shown in Table S1.

As shown in Fig. 2 (Table S1), the average gene concentrations for sulII, tetW, and intI1 were similar (~1 × 10−4 gene copies/16S rDNA copies). The tetW was the highest (1.43 × 10−4 gene copies/16S rDNA copies). The concentrations of ermF and tnpA were 5.98 × 10−6 and 4.67 × 10−5 (gene copies/16S rDNA copies), respectively (lower by one and two order of magnitudes). In ordinary one-way ANOVA, diagnostic of residuals showed data passed the Brown-Forsythe test and Anderson-Darling test (α = 0.05). One-way ANOVA showed different manure management had no significant effect on four of the five genes. Effect of different manure management practices was only found significant on tetW (p = 0.01). The Tukey test for multiple comparisons showed tetW in Compost Pile were significantly lower than Flushed Manure (adjusted p = 0.02) and Primary Lagoon (adjusted p = 0.02).

Figure 2 Copies of resistance related genes normalized to the number of bacterial 16S rRNA gene genes in different dairy manure.

X-axis labels indicate the type of dairy treatments, rectangular boxes indicate the interquartile range of the data; median value is indicated by the horizontal line inside the box. Whiskers show min to max of data. Extreme outliers (<Q1 – 3 IQ or >Q3 + 3 IQ) were removed and shown as “--” in Table 2. FM, Flushed Manure; PL, Primary Lagoon; SL, Secondary Lagoon; FP, Fresh Pile; CP, Compost Pile. Liquid samples and solid samples are separated by a red vertical line.

PCA and cluster plots for relative abundance of five genes were drawn by MetaboAnalyst 3.0 (Xia et al., 2015) as shown in Fig. 3. Relative abundance of five genes were log transformed and then normalized by median, followed by mean centering as the data scaling method. Figure 3A shows PC 1 captured 40.1% of the variation between samples, and PC 2 captured 23.4%. These two PCs captured 63.5% of the variation between the samples. The CP, FP, PL, SL, and FM groups were overlapped, which means they were not significantly different from each other. In agglomerative hierarchical cluster analysis shown in Fig. 3B, each sample began as a separate cluster and the algorithm proceeded to combine them until all samples belonged to one cluster. Results showed that PL and FM, CP and FP were similar, as they tended to cluster together. However, different manure conditions did not fall into separate clusters, indicating their ARG profiles were not significantly different from each other. As the CP, FP, PL, SL, and FM groups were overlapped in Fig. 3A and did not fall into separate clusters in Fig. 3B, it can be inferred that liquid-solid separation, lagoon system and piling process may have limited to no impacts on ARGs reductions.

Figure 3 PCA analysis.

(A) Principal Component Analysis (PCA) plot (colors representing 95% confidence regions). (B) Hierarchical clustering plot (distance measure using Euclidean, and clustering algorithm using Ward).

Co-occurrence of ARGs, MGEs, and microbial communities

Figure 4 shows the correlation network of five genes with top 50 bacterial taxa in the manure samples. Bacterial community data was used for network analysis. Range for edges was set to partial correlation values (corr. pcor) of <−0.20 or >+0.20. Red lines indicate positive correlation, while blue lines represent negative correlation. A bold line shows a p-value less than 0.05. These were three significant correlations: tnpA – sulII (corr. pcor = 0.415); intI1 – Psychrobacter (corr. pcor = 0.519); ermF – Pseudomonas (corr. pcor = 0.466).

Figure 4 Network analysis of targeted genes with bacterial communities.

Red line: positive correlation; blue line: negative correlation; bold line: p-value < 0.05.

Discussion

Prevalence and quantification of resistance related genes

The PCR results showed manure under different conditions possessed variety of ARGs and MGEs. Both traditional PCR and RT-qPCR were able to amplify DNA. RT-qPCR provided both qualitative and quantitative data by measuring the kinetics of the reaction in the exponential phase. Traditional method by agarose gels provided only qualitative results by measuring amplification products at endpoint of the PCR reaction (Parashar et al., 2006). In our study, targeted genes were screened firstly by PCR and selected gene were then quantified by RT-qPCR. It was noticed that some of genes were not detectable in PCR, and the same genes were detectable in qPCR. For example, sulII, tnpA and intI1 in PL2 were detectable in qPCR but were not detectable in PCR. This may be due to the limitation of UV visualization because some bands in agarose gels were not visible clearly under UV light. Relative abundance of intI1 in PL2 and SL5 samples was both above average in qPCR but intI1 gene in these samples was not detected in PCR.

The results showed that sulII, ermF, tnpA, and intI1 concentrations were not significantly different among five manure conditions (FP, CP, FM, PL, SL), and only one gene—tetW, was found at a significantly lower concentration in CP compared with the FM and PL. Previous studies showed various responses of ARGs to biological treatment methods such as anaerobic lagoons and composting (Zhang et al., 2021). This may be due to different experimental conditions and complex microbial ecologies involved (Pruden et al., 2013). McKinney et al. (2010) observed reductions of tet ARGs but increases of sul ARGs in anaerobic lagoons. Zhang et al. (2017) found that absolute abundances of 13 out of 14 ARGs and two integrase genes increased after 52 days of anaerobic digestion of swine manure. Sun et al. (2016) stated that four out of 10 detected ARGs declined during dairy manure anaerobic digestion under 20 °C. Storteboom et al. (2007) reported reduction of tetO but increase of tetW during horse manure composting process. Previous studies reported a higher decrease of cultivated antibiotic resistant bacteria in composting process compared to lagoon system (Wang et al., 2012). While IS6, family members of bacterial and archaeal insertion sequences are known to play crucial role in spreading antibiotic-resistance genes, overproduction inhibition phenomenon presents challenges (Muñoz-López & García-Pérez, 2010; Harmer & Hall, 2019). The overproduction and inhibition phenomenon is the property of some transposases where they display lower activity at higher concentrations; this could affect the detection of transposases as the concentration must be taken into account when evaluating impacts of these genes (Harmer & Hall, 2019; Harmer & Hall, 2020). A pitfall in the current study is that the primer used to screentransposons tnpA was specific to those previously observed in the IS6, and may have limited complete identification of all potential tnpA in the samples; a general concern with transposons is the potential role they may play on dissemination of various ARGs, and further studies would need to be conducted to specifically quantify their corroboration towards this effect.

It was noticed that average sulII, tetW, and intI1 concentrations identified in this study were lower than previous findings. As an example, McKinney et al. (2010) reported sulII and tetW of ~10−1 and 10−2 copies/16S rRNA respectively in a dairy lagoon samples in Colorado. Dungan, McKinney & Leytem (2018) reported intI1 gene copies of 10−2 /16S rRNA gene in the dairy wastewater in Idaho. Differences in ARG levels may be due to site-specific physical/chemical conditions, manure handing methods, and historical intensity of antibiotic use (He et al., 2020). However, tet and sul were reported to be the most abundant ARGs in livestock waste (He et al., 2020), which is aligned to the findings of this study.

Co-occurrence of ARGs, MGEs, and microbial communities

Integrons and transposons have been reported as responsible for the acquisition and dissemination of ARGs by HGT, which indicates that HGT could be a potential mechanism for the spread and dissemination of specific ARGs (Han et al., 2016; Sandberg & LaPara, 2016). In addition, the sulII gene was reported on a broad-host-range (BHR) plasmid RSF1010 (Rådström & Swedberg, 1988; Yau et al., 2010), and BHR plasmids are of considerable interest because they are reported not only in single bacterial species but also members of different taxonomic group, and play an important role in HGT. Existing knowledge in terms of how anaerobic processes influences ARGs is weak, particularly the effects of anaerobic digesters treating dairy manure on ARGs of manure is not well understood. Preliminary research such as a study by Huang et al. (2019) investigated the abundance of ARGs and transposase genes during anaerobic process, and authors found a reduction in ARG and transposons genes during AD of pig manure. These findings suggest that transposons genes could be correlated with ARGs. Correlations between other genes were not significant, and this may be due to resistance genes not located in integrons/transposons and non-specific selection agents in the manure (Andersson & Hughes, 2010; Di Cesare et al., 2016; McKinney et al., 2010).

While assessing the presence of ARGs, it is crucial to understand the correlation between ARGs and various microbial species. Previous studies indicate that gram-negative bacteria such as Psychrobacter and Pseudomonas, which are abundant in the environment and able to tolerate both cold and hot environment, showed correlation with ARGs and MGE. As an example, one of the species in Pseudomonas genus, Pseudomonas aeruginosa, is an opportunistic pathogen that causes infections in humans with a high mortality rate. Presence of ermF in Pseudomonas could compromise clinical treatment by MLSB antibiotics. Pseudomonas is resistant to a variety of antimicrobials due to multidrug efflux pumps, chromosomal mutations and the acquisition of resistance genes via HGT (Poole, 2011). These findings are important because the presence of ARGs in environmental bacteria have the potential to be received by human and animal pathogens through HGT, which increases the risk to public and animal healths via antimicrobial resistance (AMR) exposure.

While these results indicate the possibility of putative microorganisms hosting ARGs, additional studies are needed for strengthening the findings considering a preliminary nature of this study. A previous study by Flores-Orozco et al. (2020) identified more than 180 unique antibiotic-resistance genes in dairy manure, and changes in ARGs levels were found to be related with the shifts in the microbial community. Detailed bioinformatic analysis was conducted to evaluate the co-occurrence of microbial groups and ARGs results revealed the presence of potential microbial ARG hosts.

Many ARGs such as tetX, sul1, sul2, and tetG were dominant in various farm’s soils (Duan et al., 2019). Studies (Duan et al., 2019; Zhang et al., 2021) focused on evaluating the impacts of manure on soil’s ARGs, and anaerobic digestion process effects on the reduction of ARGs in manure suggested that cropland’s soil receiving manure showed more than two times higher ARGs compared to the soil without manure. Manure treatment process such as anaerobic digestion was found to be effective in reducing ARGs levels in manure, and changes in microbial community during anaerobic digestion resulted in reduced ARGs level (Flores-Orozco et al., 2020; Zhang et al., 2021).

One of the major limitations of this study is fewer sample numbers and samples from limited dairy farms. Further, manure management practices change from one farm to another substantially, which pose challenges in comparing the microbial and ARGs data among farms. Understanding the deviation in characteristics of ARGs and microbial community within and between farms with similar management practices can provide greater understanding of temporal changes in ARGs present in liquid manure versus solid manure.

Conclusions

In this research, we studied the prevalence of ARGs and MGEs in flushed manure, primary lagoon manure, secondary lagoon manure, fresh pile manure, and compost pile manure. Manure samples were obtained from multiple dairy farms located in Central Valley, California. Prevalence of genes varied among sample types, but all of the studied genes were detectable in different manure types. Among five genes quantified, only tetW was found at significantly lower concentration in compost pile comparing with flushed manure (adj. p = 0.02) and primary lagoon samples (adj. p = 0.02). Results of this study showed that ARGs are widely present in liquid (lagoon samples) and solid dairy farm manure (fresh and compost piles). Different manure management such as liquid-solid separation, piling, and lagoon storage may not have significant impacts on ARG and MGE reductions.

Supplemental Information

Supplemental Information 1 qPCR results for 33 dairy samples.

Click here for additional data file.

Supplemental Information 2 qPCR results for 33 dairy samples.

Click here for additional data file.

The authors thank the Division of Agriculture and Natural Resources (ANR), University of California, Davis, for providing the resources to carry out field study on dairy farms and sample collection from multiple dairy farms. The authors also thank Dr. Noelia Silva Del Rio, Dr. Alejandro Castillo, and Betsy Karle, University of California Cooperative Extension, California for their support field study and sample collection.

Additional Information and Declarations

Competing Interests

Author Contributions

Data Availability

Sharif Aly is an Academic Editor for PeerJ.

Yi Wang conceived and designed the experiments, performed the experiments, prepared figures and/or tables, authored or reviewed drafts of the paper, and approved the final draft.

Pramod Pandey conceived and designed the experiments, performed the experiments, prepared figures and/or tables, authored or reviewed drafts of the paper, and approved the final draft.

Colleen Chiu conceived and designed the experiments, performed the experiments, analyzed the data, authored or reviewed drafts of the paper, and approved the final draft.

Richard Jeannotte conceived and designed the experiments, analyzed the data, authored or reviewed drafts of the paper, and approved the final draft.

Sundaram Kuppu conceived and designed the experiments, analyzed the data, authored or reviewed drafts of the paper, and approved the final draft.

Ruihong Zhang conceived and designed the experiments, authored or reviewed drafts of the paper, and approved the final draft.

Richard Pereira conceived and designed the experiments, authored or reviewed drafts of the paper, and approved the final draft.

Bart C. Weimer conceived and designed the experiments, authored or reviewed drafts of the paper, and approved the final draft.

Nitin Nitin conceived and designed the experiments, authored or reviewed drafts of the paper, and approved the final draft.

Sharif S. Aly conceived and designed the experiments, authored or reviewed drafts of the paper, and approved the final draft.

The following information was supplied regarding data availability:

The raw data is available in the Supplementary Files.

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
