# Peer review of "Quantification of antibiotic resistance genes and mobile genetic in dairy manure"

_PeerJ, doi:10.7717/peerj.12408_

## Round 0.1 · original submission · Major Revisions

Dear Author, your manuscript has been reviewed and the comments from both reviewers you can find attached. Both the reviewers found your work interesting but requiring a major revision. I side with their judgement. Please refer to the comments and prepare the revised version accordingly to the suggestions.

·

Basic reporting

In general, the manuscript is well-structured, and the literature is well referenced.

There are a few sections where the language should be improved, including:
• Lines 90-91.
• Lines 97-99.
• Lines 117-126
• Line 159

Table 3 and Figure 2 show the same results; therefore, they are redundant. It would be better to keep only the figure in the manuscript and provide the table as supplementary material.

Besides, table 43 is too crowded and hard to read. It has some values that do not make sense such as 4.18E+00, or 0.00E+00. Again, I consider it could be removed from the manuscript and provided as supplementary material.

Experimental design

In general, I consider this study had a solid and well-planned experimental design.

However, my main concern is that the analysis is based on the average of the samples from different farms, which, in my opinion, could mask the real fate of the ARG on the different stages of the manure management system.

Validity of the findings

This study found not very significant differences in ARG levels in different stages of a manure management system. The results are interesting since most of the studies have found different ARG levels under different conditions, which is generally attributed to the different microbial populations present in each stage. It would be useful to include some data about the general microbial composition of the different sampling points to strengthen these results.

My major concern is that these small differences in ARG levels on the different stages could be caused by averaging out the results of the different farms sampled in this study. If possible, it would be useful to group and present the data by farm and stage and then evaluate if there is a difference within each farm’s samples (along the process). That is the only way to conclude that the different stages/conditions of the manure in operating farms affect or not, the fate of ARGs since the condition on each farm could be very different.

It would be useful to include some more discussion on the putative microorganisms hosting ARGs. There are plenty of studies that have found interesting correlations that fit with the results reported in this study. This could strengthen the manuscript.

Additional comments

There are some other minor corrections suggested to the manuscript:

It would be helpful to include a short description of the reason for the high concentration of antibiotics in manures (low absorption in the body) to make a better transition between the paragraph ending on line 55 and the new one starting in line 56
.
Line 58: Dairy cattle are considered potential mediators, reservoirs, and disseminators of resistant bacteria and/or ARGs. Shouldn’t it be dairy/cattle manures?

Line 82: …the understanding of regulating ARG in the process is still unknown. It is not clear what the authors are trying to express.

Line 87: lower instead of lowest.

Line 92-93: Most manure conditions were simulated at the bench scale, and only few analyzed commonly used practices in dairy farms. This sentence is not aligned with what is described in the paragraph, which states the finding on ARGs present in manures of different operating farms.

Lines 106-116. I am not sure if this paragraph provides valuable information for the manuscript since it mostly describes the limitation of a culture-based approach for the identification of antibiotic-resistant bacteria. It would be better to describe the advantage of using qPCR or a justification of why using qPCR to do this and not the use of culture-based or metagenomic approaches.

Lines 117-118: The goal of this study was to investigate the fate of ARGs and MGEs under dairy farm manure management. Include the fact that the manure management is a flushed system.
Lines 118-120: I consider that the author should refer to the “different manure management conditions” as different stages of the manure in the flushed system or something similar so the readers could quickly and easily associate them with the system they studied.

Line 130-131: Sample information is described elsewhere and 16S rRNA gene sequencing results were published. This sentence does not make sense in this place.

Line 132: have or account for instead of has.

Line 144: L instead of liter.

Lines 153-155: Liquid samples with clear-to-low turbidity were processed by the MO BIO PowerWater® kit, and 10 - 200 mL of each was filtered through a Millipore filter (0.45-μm pore size). Just wondering, could this filtration have retained some microbial biomass and affect the DNA isolation?

Line 226-227: Four manure management groups, FP, FM, PL, and SL, had similar positive percentages of gene types. You should refer to the manure samples (conditions or stages of the flushed system) consistently through the whole document to avoid confusion.

Line 241-242: DNA templates for qPCR were the same batch of extractions as PCR. Include reference to Table 1.

Line 325-326: This study showed that tnpA and sulII abundance were significantly correlated, which indicates that the sulII was possibly related to the HGT by transposons. This sentence is wrong or there is not data supporting it. It is true that sulII was correlated to tnpA but there is not evidence in this study of gene transfer of any sort. Please support this sentence or modify it.

Lines 340-343: Although current animal waste treatment systems in dairy farms are not specifically designed to remove ARGs, it is important to understand the potential impacts of existing manure management practices on removal of ARGs. ARGs are considered as an environmental contaminant, which may adversely impact human health. This part should not be part of the conclusions, it does not fit here and it is not necessary.

Reviewer 2 ·

Basic reporting

I found the introduction a bit too long, with several examples for each idea the authors want to highlight. I think this could be shortened. If all the references are too be maintained, more general sentences could be written.

Figures S1-S6 are not provided by the authors.

In Table 2, it is not easy too the eye to understand what the “average column” refers to.

Experimental design

Animal manure is often used as organic fertilizer as an alternative to chemical fertilizers for arable soils of low fertility, and antibiotic residues, bacteria and antibiotic resistant genes are frequently detected in manure and manure amended soils. As the health of people is closely connected to the health of animals and our shared environment (One Health), it is important to explore antimicrobial resistance genes content of manure and the optimal management practices to reduce that content, once resistance may compromise the effectiveness of antibiotic treatment of bacterial infections. The authors show that different management practices don’t significantly impact the amount of ARGs in manure, highlighting the need for improved practices.

Some information is missing in the methods. For instance, detection of intI and tnpA genes is not mentioned anywhere in the text. Additionally, the primers listed in Table 1 for intI do not present the original primers reference, which is Barlow et al. 2004 AAC. Nothing is also mentioned about the fact that the tnpA primers only target the transposase of the IS6 group, not all tnpA in general; this changes the way results must be interpreted.

Validity of the findings

No comment

Additional comments

The authors screen for the presence of two genes that are usually part of a MGE, but are not a complete MGE by themselves. For instance, the intI gene is not a MGE; it is usually part of class 1 integrons. And detecting the intI1 gene does not ensure that the complete integron is present. This should be clarified; intI1 may be used as a marker or an indicator of a MGE presence, but it is not a MGE. An evidence supporting this observation is the fact that the authors did not find a correlation between intI and sulI, which are both part of the conserved segments of class 1 integrons.

Another thing that should be clarified is that HGT occurs by conjugation, transduction or natural transformation. HGT is not mediated by transposons (unless they are conjugative) nor integrons (lines 67-68, 317-318, 326, 350). Acquisition of naked DNA by natural transformation may also be relevant in the context of ARGs dissemination, especially in this type of environment.

The sentence in lines 321-2 needs to be re-written. Plasmid RSF1010 is not integrated into transposons; rather, the transposons mentioned in the reference (Cain et al 2010) carry genes that are derived from that plasmid, which is different.

---

## Round 0.2 · Major Revisions

Dear Author, please find attached the reviews of your manuscript. Both reviewers pointed value of your work but also both suggested revision. I trust their judgment. Please prepare your revision accordingly to their suggestions. With kind regards, Associate Editor

·

Basic reporting

In general, the manuscript is well-structured, and the literature is well referenced.
There are a few sections where the language should be improved and/or contains typos, for example:
Line 50: causND ing
Line 58: manures instead of manure.
Line 93: haveL

Thorough proofreading is recommended.

Figures and tables are well presented now.

Experimental design

In general, I consider this study had a solid and well-planned experimental design. The authors have addressed the recommendations provided in the first review.

Validity of the findings

The study provides interesting insight into the ARG evolution in manure treatments. Most of the previous concerns were addressed by the authors.


It would be useful to include some more discussion on the putative microorganisms hosting ARGs. For instance, Line-297-299: “Network analysis indicates intI1 and Psychrobacter, ermF and Pseudomonas, were significantly correlated (p < 0.05). This suggests Psychrobacter was potential hosts of intI1, and abundance of ermF could be attributable to the presence of Pseudomonas.” This paragraph is short and should be expanded to include some of the potential microbial species hosting ARG reported in previous studies and briefly discuss if the finding of this study agrees or not with them. A few studies provide interesting results on this, for example:
Duan, M., Gu, J., Wang, X., Li, Y., Zhang, R., Hu, T., & Zhou, B. (2019). Factors that affect the occurrence and distribution of antibiotic resistance genes in soils from livestock and poultry farms. Ecotoxicology and Environmental Safety, 180, 114–122. https://doi.org/10.1016/j.ecoenv.2019.05.005
Flores-Orozco, D., Patidar, R., Levin, D. B., Sparling, R., Kumar, A., & Çiçek, N. (2020). Effect of mesophilic anaerobic digestion on the resistome profile of dairy manure. Bioresource Technology, 315, 123889. https://doi.org/10.1016/j.biortech.2020.123889
Zhang, Q., Xu, J., Wang, X., Zhu, W., Pang, X., & Zhao, J. (2021). Changes and distributions of antibiotic resistance genes in liquid and solid fractions in mesophilic and thermophilic anaerobic digestion of dairy manure. Bioresource Technology, 320, 124372. https://doi.org/10.1016/j.biortech.2020.124372

Additional comments

There are some other minor corrections suggested to the manuscript:
Line 53-54: “The high concentrations of antibiotics in excreted manure is potentially due to low absorption in the cattle body”. A reference for this is missing. You can use (Jjemba, 2002),
• Jjemba, P. K. (2002). The potential impact of veterinary and human therapeutic agents in manure and biosolids on plants grown on arable land: A review. Agriculture, Ecosystems and Environment, 93, 267–278. https://doi.org/10.1016/S0167-8809(01)00350-4

Line 277-279: “Previous studies showed various responses of ARGs to biological conditions such as anaerobic lagoons and composting”. Include a citation here.

Line 300 – 301: “In general, ARGs are persistent in the environment, and it is well established that many putative microorganisms hosts ARGs”. I feel this sentence does not fit in this section, please remove.

Reviewer 2 ·

Basic reporting

Some references are lacking, especially in newly added sentences.

Experimental design

Authors included the original reference of intI primers, but kept the previous reference. Why?
Although the authors said they included a sentence regarding the limitation of tnpA and IS6, they did not do it.

Validity of the findings

Some limitations of the study were not stated.

Additional comments

First of all, it would have been helpful if the authors have included the lines where changes were done in the responses to the reviewers. It was difficult to find where were the changes they said they did it.

In my opinion, the authors did not put effort in improving the manuscript, but rather copied comments from both reviewers, without including supporting references. I think they need to re-read the comments from the first review and really improve the manuscript.

Additionally, the authors said they deleted some information, but they did not. For instance, it is still written that “The plasmid was also found to be integrated into transposons” (line 306), which is wrong.

---

## Round 0.3 · Major Revisions

Dear Author,

Please find comments from the Reviewers attached. Please update your manuscript accordingly. Good luck.

With kind regards,
Robert Czajkowski (Editor)

·

Basic reporting

Thorough proofreading is still required, especially in the content added in this new revision.

Line 323-331: “Elevated levels of ARGs and IntI1 were found in many farm samples such as chicken farms, pig farms, and beef cattle farms”. This sentence requires a citation.

Experimental design

Line 151-155. “While this study evaluated the presence of two genes that are usually part of a MGE, it is important to note that they are not complete MGE by themselves. For instance, the intI gene is not a MGE; it is usually part of class 1 integrons, and detecting the intI1 gene does not ensure that the complete integron is present. The intI1 may be used as a marker or an indicator of a MGE presence, however, it is not a MGE”. I feel this part does not really fit in this section. It sounds more like a discussion of the limitations rather than methodology.

Validity of the findings

The quality of the manuscript has certainly improved. However, some minors corrections are required, especially in the new content.

Line 285-290. “While IS6, family members of bacterial and archaeal insertion sequences are known to play crucial role in spreading antibiotic-resistance genes, overproduction and inhibition phenomenon presents challenges (Munoz-Lopez and Garcia-Perez, 2010; Harmer and Hall, 2019). The use of transposons, tnpA, has disadvantages due to the generally nonspecific nature of target site selection (Fickman and Dyda, 2016).”
These new sentences should be developed a bit more and framed better. For example, the part “…overproduction and inhibition phenomenon presents challenges” should include some details in the challenges. I imagine that the idea was to say that the overproduction and inhibition of IS6 do not necessarily mean an increase or decrease in HGT.
The part “The use of transposons, tnpA, has disadvantages due to the generally nonspecific nature of target site selection” is only focused on the limitation and does not provide an actual discussion on the meaning of the finding. The author could mention that even though the presence of tnpA does not necessarily mean an increase ARG spread, it could suggest a greater risk of HGT, or some similar ideas.

Line 323-340 : “Elevated levels of ARGs and IntI1 were found in many farm samples such as chicken farms, pig farms, and beef cattle farms. Further many ARGs such as tetX, sul1, sul2, and tetG were dominant in farm soils (Duan et al., 2019). Studies (Duan et al.,2019; Zhang et al., 2021) focused on evaluating the impacts of manure on soil’s ARGs, and …”. I consider this part does not belong to the paragraph. It could stand as a separate paragraph.


Line 338-340. “In addition, this will help in developing improved manure management practices, which are capable of unwanted microbial population and ARGs in manure.” This sentence needs to be corrected. I assumed the intention was to say that manure management practices are capable of reducing microbial populations potentially harboring ARGs.

Reviewer 2 ·

Basic reporting

No comment

Experimental design

No comment

Validity of the findings

Misunderstanding of some concepts limit the validity of the discussion. More details are added below.

Additional comments

The authors overlook my comments about fully copied paragraphs from reviewers suggestions, including lack of references (for example, lines 93-99, 151-155 and 301-2). The idea present in lines 93-99 is already partially included in lines 53-62; the ideas should be included only in one place. Lines 151-155 were included in material and methods, when they are more a limitation of the manuscript as it was discussed. Regarding transposons, lines 300-1 have the opposed meaning of lines 301-3. The same idea is here: “Network analysis showed that sulII was significantly correlated with HGT by transposase gene” (lines 349-350) cannot be said, as tnpA does not promote intercellular movement by HGT, it is only able to promote intracellular movement.

Additionally, the limitation of the use of primers for tnpA of IS6 group is not included. The authors mention IS6 once in the discussion (line 285), without a previous explanation before. The readers won’t understand that IS6 is mentioned because the primers the authors use for tnpA detection are specific to this family. Furthermore, the sentence used (from Hickman and Dyda) does not apply here; the nonspecificity is related with the target site sequence. In this study, the authors amplify the transposase gene by PCR, which is different and specific for this IS family.

---

## Round 0.4 · Minor Revisions

Dear Authors, as suggested by the Reviewer, your manuscript requires an additional round of revisions. Good luck with addressing all comments and suggestions.

·

Basic reporting

Some typos:
Line 35: “…which are considered to be responsible for the dissemination of ARGs in environment”. Should be “… in the environment”.
Line: 46: “concernt” instead of concern.
Line 104: there is a dot at the beginning of the sentence.
Line 111: “under” instead of under.
Line 201: “useing” instead of using.
Line 348: “… limited dairy farms,. Further…”
Line 350: “an” instead of and

Experimental design

NA

Validity of the findings

I still feel that the discussion on the cooccurrence of ARGs, MGEs, and microbial communities is weak and should be structured better because it is a little hard to follow. For instance, it should start stating the main correlations found in the study and then proceed with the discussion.
Also, the following sentences need to be frame better in the context of this study since they seem disconnected:
• Line 318-319: “The sulII gene was reported on a broad-host plasmid RSF1010 (Rådström and Swedberg 1988; Yau et al. 2010)”. This sentence needs to be aligned with the previous sentence on the importance of determining ARG transfer mechanisms.
• Line 319-320: “Huang et al. (2019) reported abundance of ARGs and transposase genes, which were decreased during anaerobic digestion of swine manure”. It would be better to say that Huang et al. (2019) found a reduction in ARG and transposase genes levels during AD of pig manure, which suggested that transposase genes could be correlated with ARG or something of the sort linked to your findings.
• Line 324: the discussion on the microbial species could stand as a separate paragraph. It could start highlighting that Psychrobacter and Pseudomonas were correlated with ARG and MGE. Then describe why this finding is important, mainly what is already there but framed a little better.

---

## Round 0.5 · accepted · Accept

Dear Author, I am happy to accept your manuscript for publication in PeerJ journal.
with kind regards,
Editor